# LANGUAGE MODELS INVERSELY SCALE ON PIECEWISE FUNCTION EVALUATION WITH BIASED EXAMPLES

**Jordan Juravsky,**\* **Bradley Brown,**\* **Atif Mahmud,**\* **Ryan Ehrlich,**\* **Wais Shahbaz**\*
University of Waterloo
{jordanjuravsky,bradley19brown,atifmahmud101,ryansehrlich,wais.shahbaz}@gmail.com

## ABSTRACT

We investigate whether language models (LMs) can be misled by providing them with factually correct, but unrepresentative/biased examples, in the context of integer-to-integer piecewise functions. Given the definition of a piecewise function and several examples of the function's evaluation, we instruct LMs to apply the function to a new input. We assess LMs on two variants of this task: one where the example function evaluations are evenly distributed across both branches of the function, and one where all of the examples exercise one branch of the input and the target input exercises the other branch. We observe that model performance positively scales with model size only when examples are balanced, and that performance inversely scales with size when the examples are misrepresentative.

## 1 INTRODUCTION

Large language models (LLMs) have recently been widely-deployed to hundreds of millions of users (OpenAI, 2022; Mehdi, 2023). A standard way to interface with these models is by using prompts, where the user provides a description of the problem to solve, often alongside examples of similar problems being solved (Brown et al., 2020). This few-shot approach has led to state-of-the-art performance on a variety of natural language processing benchmarks (Chowdhery et al., 2022). Moreover, this technique has been shown to scale well with model size: larger models regularly outperform smaller models when provided with the same task/prompt (Wei et al., 2022). Recent work has provided counterexamples to this trend, discovering "inverse scaling" tasks where performance is negatively correlated with model size (Perez et al., 2022a; Perez & McKenzie, 2022; McKenzie et al., 2022; Perez et al., 2022b).

In this work, we introduce another such inverse scaling task, which uses the evaluation of piecewise functions as a mechanism for exploring the interaction between a prompt's problem description and the examples provided in it. In particular, we show that when the provided examples for this task cover a diverse set of problem inputs, model performance scales positively with size. However, when all of the examples are drawn from a subset of the input space, and the desired problem input is drawn from outside this set, we observe inverse scaling. While in this domain one can determine by inspection that the examples are unrepresentative, we use this toy task to highlight a failure case of language models that may appear in scenarios where the example bias is less obvious.

## 2 TASK FORMULATION

We assess the OPT (Zhang et al., 2022) and GPT3 (Brown et al., 2020) language model (LM) families on a dataset of piecewise function evaluation problems. Each problem in the dataset references a randomly generated function $f : \mathbb{Z} \to \mathbb{Z}$, defined as:

$$f(x) = \begin{cases} g(x) & \text{if } \rho(x) \\ h(x) & \text{otherwise} \end{cases} \tag{1}$$

---

\*equal contribution, author order randomized

- where $\rho(x) = \rho_1(x) \land ... \land \rho_n(x)$ is a conjunction of predicates on the input integer $x$ and two function branches $h : \mathbb{Z} \to \mathbb{Z}, g : \mathbb{Z} \to \mathbb{Z}$ are arithmetic functions that either add, subtract, or multiply $x$ by a constant.

The prompt for each problem in our dataset contains three components: the definition of $f$ in natural language, several example evaluations of $f$, and a request to evaluate $f$ on a new input $q \in \mathbb{Z}$ (the "target input"). Each problem is formulated as a two-option classification task, where the classes are $[h(q), g(q)]$ and $f(q)$ is the correct answer. See Appendix C for examples of prompts, and Appendix B for more details on our dataset generation procedure.

We evaluate LMs on two variants of this task. In the "balanced" variant, the provided example function evaluations are evenly distributed across the two branches of $f$ (i.e. for half of the examples, the predicate conjunction is true, and for the other half it is false). In the "biased" variant of our task, we deliberately select examples so that all of them exercise the same branch of $f$, while $q$ exercises the opposite branch. We emphasize that in both variants of this task, the provided function evaluations are correct, and only differ in the inputs that were chosen. Moreover, we highlight these example function evaluations are not required information for solving the problem of evaluating the target input properly, since the function definition itself is included in the prompt.

## 3 RESULTS

Our key results are summarized in Figure 1. When evaluating LMs on problems with balanced examples, we observe normal scaling behaviour, with accuracy increasing with model size. However, on the problems where the provided examples are biased to the branch that is opposite to the target input, we observe strong inverse scaling behaviour. The largest (175B parameter) models that we evaluate perform over 15% worse than random guessing. This is surprising, since one might expect the higher capacity LMs to understand the function definition better than smaller models, and therefore rely less on the provided examples, regardless of whether they are biased or not. We ablate these experiments over different dataset parameters, and provide the results in Appendix D.

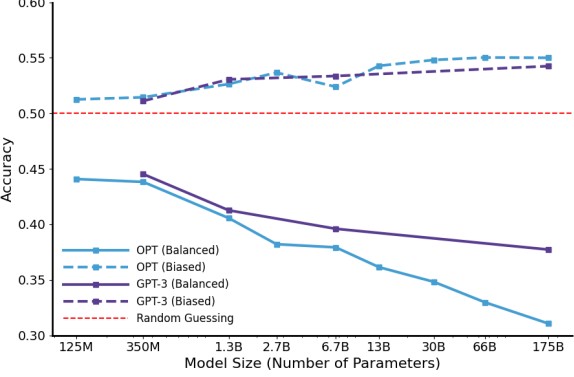

Figure 1: We observe that accuracy positively scales with model size when using balanced examples, while accuracy inversely scales with size when using biased examples.

## 4 CONCLUSION

In this work we introduced piecewise function evaluation as a mechanism for assessing how language models of different sizes behave when provided with correct, but unrepresentative examples. We showed that when provided with a biased set of function evaluations, LLMs are prone to base their prediction on the few-shot examples rather than the definition of the function. Additionally, we showed that this failure mode gets more severe as the model gets larger. We are interested in future work that discovers additional domains where LLMs can be misled by factually correct, but misrepresentative examples.

URM STATEMENT

All authors (Jordan Juravsky, Bradley Brown, Atif Mahmud, Ryan Ehrlich and Wais Shahbaz) meet the URM criteria of ICLR 2023 Tiny Papers Track.

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

## A    RELATED WORK

**Language Models:** Language models (LMs) have significantly advanced the state-of-the-art in many natural language processing tasks. When deep learning architectures such as transformers (Vaswani et al., 2017) are trained with self-supervised objectives (such as next-token or masked token prediction) on large datasets of text extracted from the Internet (Gao et al., 2021; Merity et al., 2016), the resulting models are able to develop powerful internal representations for natural language (Devlin et al., 2018; Liu et al., 2019; Raffel et al., 2019). These pre-trained LMs can either be directly applied or fine-tuned to solve a diverse set of tasks (Wang et al., 2018; 2019; Srivastava et al., 2022).

**Scaling and Inverse Scaling:** Strong scaling properties have been established for language models: that is, as the size of language model architectures and the datasets that they are trained on increase, so too does their performance on many tasks (Brown et al., 2020; Rae et al., 2021; Chowdhery et al., 2022). Many of these *large* language models (LLMs) exhibit emergent abilities (Wei et al., 2022), demonstrating a sharp increase in performance on many tasks that smaller models struggle with. Notably, recent work (Perez et al., 2022b), including submissions to the Inverse Scaling Competition (Perez et al., 2022a; Perez & McKenzie, 2022; McKenzie et al., 2022), has found counterexamples to this scaling trend, identifying tasks where larger model perform progressively worse ("inverse scaling tasks").

**Zero-shot and Few-shot Prompting:** An important characteristic of many language models is that they can be applied to novel domains without fine-tuning, instead only requiring a description of the problem in the form of a prompt (Radford et al., 2019). Relevant to our work, LLM performance on these downstream problems can often be significantly improved by providing the models with examples of solved similar tasks in the prompt (i.e. few-shot examples) (Brown et al., 2020).

## B    TASK CONSTRUCTION DETAILS

### B.1    GENERATING PIECEWISE FUNCTIONS

Each problem that we provide to LMs is centered on a randomly generated function $f : \mathbb{Z} \to \mathbb{Z}$ that follows the structure:

$$f(x) = \begin{cases} g(x) & \text{if } \rho(x) \\ h(x) & \text{otherwise} \end{cases} \tag{2}$$

In this expression:

- $\rho(x) = \rho_1(x) \wedge ... \wedge \rho_n(x)$ is a conjunction of predicates on the input integer $x$ (functions $\rho_i : \mathbb{Z} \to \{\text{True}, \text{False}\}$). Each predicate is sampled uniformly at random from the options:
    - "$x$ is even"
    - "$x$ is odd"
    - "$x$ is prime"
    - "$x$ is not prime"
    - "$x$ is less than $K$" for some $K \in \{2, 3, ..., 100\}$ sampled uniformly at random.
    - "$x$ is greater than $K$" for some $K \in \{2, 3, ..., 100\}$ sampled uniformly at random.
    - "$x$ is a multiple of $K$" for some $K \in \{2, 3, 4, 5\}$ sampled uniformly at random.
- The two function branches $h : \mathbb{Z} \to \mathbb{Z}, g : \mathbb{Z} \to \mathbb{Z}$ are arithmetic functions sampled uniformly at random from the options:
    - $q(x) = x + K$ for some $K \in \{2, 3, ..., 100\}$ sampled uniformly at random.
    - $q(x) = x - K$ for some $K \in \{2, 3, ..., 100\}$ sampled uniformly at random.
    - $q(x) = x * K$ for some $K \in \{2, 3, 4, 5\}$ sampled uniformly at random.

When generating candidate piecewise functions, if we are unable to find examples $x \in \{2, 3, ..., 100\}$ that exercise both branches of the function, we discard the function and generate another. For example, if we randomly generate the predicate conjunction "$x$ is even and $x$ is odd", then no inputs can satisfy this conjunction, causing us to discard the function.

## B.2 EXAMPLE AND PROMPT GENERATION

The key hyperparameters used when generating each problem in our dataset are:

- the number of predicates used in the conjunction.
- the number of example function evaluations provided.
- the distribution of the provided examples (balanced or biased).

When generating balanced examples, we iteratively pick input values that alternate the branch of the function they exercise (we randomly pick which function branch the first example should follow). To select the target input $q$, we first randomly pick which branch of the function $q$ should follow, and then iterate over the numbers $\{2, 3, ..., 100\}$ in a random order until we find a satisfying value. This process ensures that across the problems in our dataset, the number of target inputs that exercise each branch of the piecewise function is approximately equal.

We consider two different approaches to generating biased examples and target values: an "easy" mode and a "hard" mode. When generating biased examples in easy mode, the examples and target inputs satisfy predicates in an "all-or-nothing" fashion. Specifically, prompts in easy mode fall into two cases:

- In the first case, every example evaluation satisfies every predicate, while the target input fails every predicate (for example, if the function condition is that a number must be greater than 11 and be even, all of the examples will be even numbers larger than 11, while the target input is an odd number less than or equal to 11). This causes every example input to satisfy the $g(x)$ branch of the piecewise function, while the target input exercises the $h(x)$ branch.
- The second category of prompts in easy mode is the converse of the first case, where every example fails every predicate in the piecewise function condition, while the target input satisfies every predicate.

We balance these two cases in our dataset, so that the target values exercise the $h(x)$ and $g(x)$ branches of piecewise functions are exercised with equal frequency. See Appendix C.2 for an example of a prompt with biased examples generated with easy mode.

When generating biased examples and target values using hard mode, we use more subtle biases. This mode relies on the fact that the condition $\rho$ of each piecewise function is a conjunction of predicates, and therefore the entire conjunction evaluates to false if any single predicate evaluates to false. The prompts in the hard version of our dataset rely on this fact and, like the easy mode, fall into one of two categories:

- In the first case, all of the example inputs satisfy every predicate, while the example input satisfies every predicate except for one. Since all but one of the predicates evaluate to true for the target input, this construction makes it easier for the model to misclassify the target input as satisfying the predicate conjunction and following the $h(x)$ branch that all of the provided examples exercise.
- Like with easy mode, the second category of prompts is converse of the first case, where all example inputs have all but one predicate evaluating to true (putting the examples in the function's $h(x)$ case), and the target inputs satisfy all predicates (putting it in the $g(x)$ case).

See Appendix C.3 for an example of a prompt with biased examples generated with hard mode. Furthermore, see Figure 2 for an ablation over the performance of easy and hard datasets.

### B.3 DATASET GENERATION

The final dataset that we use to produce Figure 1 is created by sweeping over hyperparameters, generating 150[1] problems for every configuration, and pooling all of these problems together. We specifically sweep over predicate conjunction sizes in {2,3} and the number of examples per prompt in {2,4}. We also sweep over three different variations of our prompt format (see examples of each version in Appendix B.4). When generating the biased example results, we also sweep over easy mode and hard mode.

Lastly, we also sweep over formulating our problems as 0-shot and 2-shot. In our task, the number of shots is distinct from the number of example evaluations provided per function, and instead refers to the number of solved function evaluation problems that we provide in the prompt. For example, a 2-shot formulation of our task means that we first provide the model with two separate questions and answers involving piecewise functions (each of which contains their own function definition, evaluation examples, target input, and answer), before providing a third question involving a third piecewise function that we ask the model to solve. Notably, the function evaluation examples in our few shot prompts are always balanced, regardless of whether the examples in the true problem are balanced or biased. The motivation behind this choice is that if we used few-shot examples that were biased in the same way as our real problem, the model could simply learn the pattern in the biases (e.g. the model could observe that all of the examples exercise one branch of the function while the target input exercises the other, and simply mimic that pattern without understanding why an input corresponds to a branch of the function).

We ablate the effects of the hyperparameters we sweep over in Appendix D.

### B.4 PROMPT VARIATIONS

We use three different variations of our prompt wording - an example of each is given below:

**Example Prompt Variant 1**

> "Let's define the piecewise function f(x) as follows: if x is a multiple of 3 and x is not prime and x is even, then f(x) = x - 42. Otherwise, f(x) = x * 5. For example: f(30) = -12, f(100) = 500, f(12) = -30, f(85) = 425. What is f(63)? Answer: f(63) ="

**Example Prompt Variant 2**

> "Setup: Consider the following piecewise function, f(x). For an input value x, if x is a multiple of 3 and x is not prime and x is even, then we define f(x) = x - 42. Otherwise, we define f(x) = x * 5. For example: f(30) = -12, f(100) = 500, f(12) = -30, f(85) = 425. Question: What is f(63)? Answer: f(63) ="

**Example Prompt Variant 3**

> "Let's create a piecewise function, f(x), that maps integers to integers. If an integer x satisfies the conditions that x is a multiple of 3 and x is not prime and x is even, then our function maps f(x) = x - 42. Otherwise, our function maps f(x) = x * 5. For example: f(30) = -12, f(100) = 500, f(12) = -30, f(85) = 425. Question: What is f(63)? Answer: f(63) ="

---

[1]when evaluating OPT-175B and GPT3-davinci, we only use 50 examples per configuration - this still yields a dataset of several thousand examples in total

## C    PROBLEM EXAMPLES

We colour the examples below using the following legend:

- ◻ Predicates
- ◻ Examples
- ◻ Function branch when $\rho(x)$ = True (all predicates satisfied)
- ◻ Function branch when $\rho(x)$ = False (at least one predicate unsatisfied)

### C.1    BALANCED EXAMPLES

"Let's define the piecewise function f(x) as follows: if x is even and x is greater than 49 and x is less than 89, then f(x) = x + 37. Otherwise, f(x) = x - 60.

For example: f(82) = 119, f(91) = 31, f(68) = 105, f(21) = -39.

What is f(80)?
Answer: f(80) ="

We see that this example has a conjunction containing three predicates, and provides four example function evaluations before asking the model to evaluate the function on the target input 80. Note that the provided examples are equally balanced between both branches of the function.

### C.2    BIASED EXAMPLES - EASY MODE

"Let's define the piecewise function f(x) as follows: if x is less than 20 and x is not prime and x is even, then f(x) = x * 3. Otherwise, f(x) = x - 19.

For example: f(97) = 78, f(41) = 22.

What is f(12)?
Answer: f(12) ="

In this problem which generates biased examples using easy mode, we see that each of the example evaluation inputs fail every predicate, while the target input satisfies every predicate.

### C.3    BIASED EXAMPLES - HARD MODE

Let's define the piecewise function f(x) as follows: if x is a multiple of 5 and x is greater than 47, then f(x) = x - 58. Otherwise, f(x) = x * 4.

For example: f(80) = 22, f(70) = 12.

What is f(48)?
Answer: f(48) =

In this problem which generates biased examples using hard mode, we see that each of the example evaluation inputs satisfies every predicate while the target input does not satisfy exactly 1 predicate.

# D    ABLATIONS

In this section, we reproduce the experiments ran to produce Figure 1, ablating over different task hyperparameters. For a given hyperparameter that we ablate over, we sweep over the remaining hyperparameters described in Appendix B.

## D.1    EASY VS. HARD MODE

Below we see the accuracy curves for the easy and hard versions of our dataset (as defined in Appendix B.2). We note that although inverse scaling occurs for both datasets, all models obtain worse performance and the slope is steeper for the hard version of the dataset.

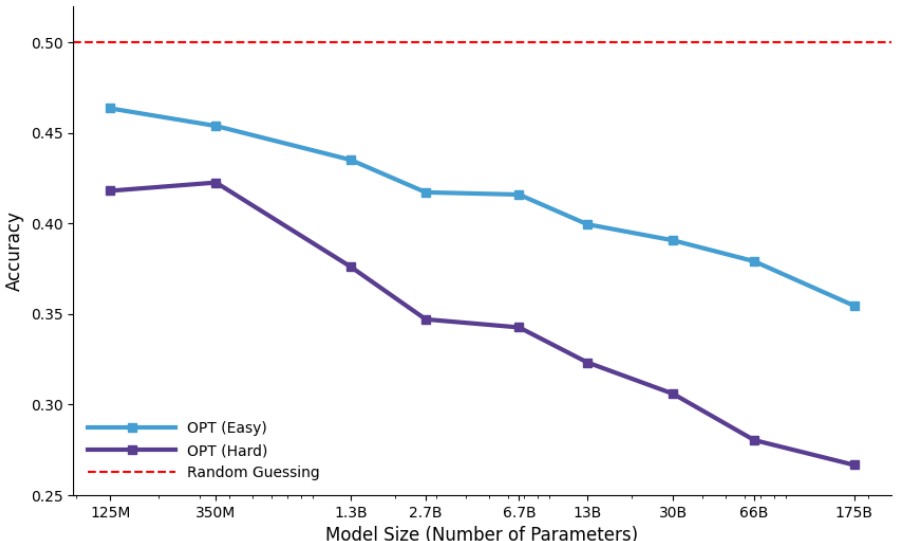

Figure 2: Classification Accuracy Scaling Plot for Easy and Hard versions of the dataset.

## D.2 NUMBER OF EXAMPLE FUNCTION EVALUATIONS

Below we measure accuracy when ablating over datasets containing different numbers of biased few-shot examples, including using no examples at all. We observe that when no example function evaluations are provided, all but the largest models possess accuracies slightly above that of random guessing. As shown in Figure 1, we observe much stronger inverse scaling when biased examples are provided.

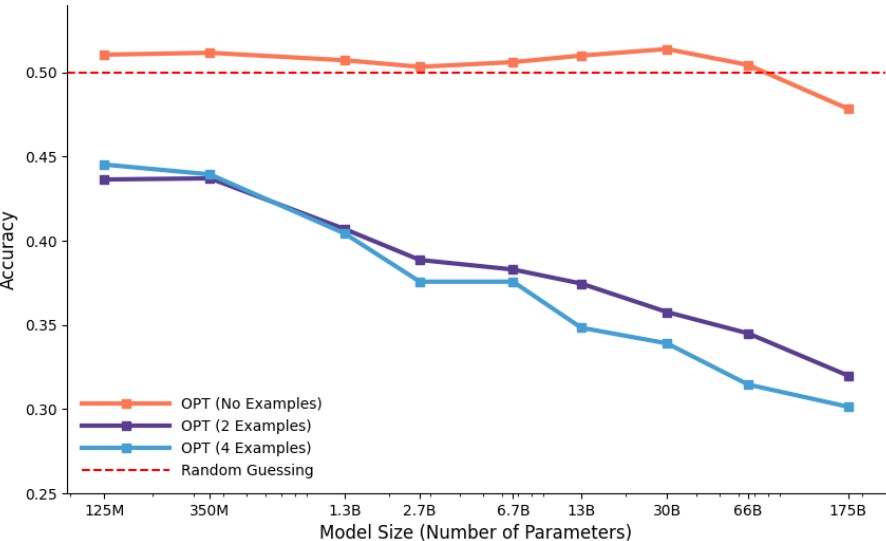

Figure 3: Classification Accuracy Scaling Plot for a Different Number of Few-Shot Function Evaluations

### D.3 SIZE OF PREDICATE CONJUNCTION

Below we measure accuracy curves when ablating over datasets whose problems have differently sized predicate conjunctions.

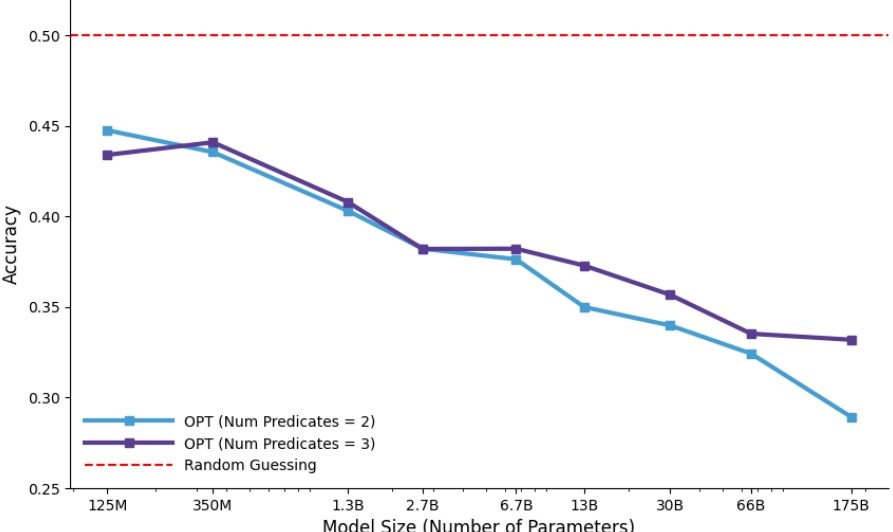

Figure 4: Classification Accuracy Scaling Plot for a Different Number of Predicates

