# OpenReview forum: "Language Models Inversely Scale on Piecewise Function Evaluation with Biased Examples"
_ICLR.cc/2023/TinyPapers — Submitted to Tiny Papers @ ICLR 2023_

### Official Review · Reviewer_9pXy · 2023-03-30

**Confidence:** 4

**Summary Of Contributions:**

The paper proposes an evaluation of LLM with few-shot examples that are misrepresentative of the target input and shows that in the context of integer-to-integer piecewise functions, the performance inversely scales with model size.

**Rating:**

Clear, Correct, and Reproducible (CCR): a submission which meets the reviewing criteria

**Strengths And Weaknesses:**

Strengths:
1. The task is well-defined.
2. The experiments with both OPT and GPT models support the claim.

Weakness:
1. It is unclear whether the results are due to the model's misunderstanding of the task definition or failure to understand which function to use.

Question: Will chain-of-thought style reasoning where the model first identifies which piecewise function to use and then generate the output might scale differently?

**Suggested Changes:**

A lot of details are included in the appendix which is crucial to understand the experimental setup. I think it would be super helpful to include one example prompt (B.4) in the main paper.

---

### Official Review · Reviewer_ABWD · 2023-04-01

**Confidence:** 3

**Summary Of Contributions:**

The paper discusses the problem of evaluating a piece-wise function using large language models. It presents a phenomena where adding biased examples actually decreases the function evaluation accuracy.

**Rating:**

Great Start (GS): a submission which meets some of the reviewing criteria but has room for improvement

**Strengths And Weaknesses:**

Strengths

* This work investigates an interesting problem in problem-solving/reasoning capabilities of LLMs. Domains where LLMs are not very accurate or non-intuitive behaviors are very important to find and this paper does a good job of presenting such a domain.

* The experiments regarding easy/hard problems and adding more negative examples in the appendix are also interesting. While I understand, they may be beyond the scope of the size requirements for this workshop, a way to concisely present these results together will make the paper stronger.

Improvement scopes

* I am not convince about the conclusion regarding balanced and biased prompts. For example, one case of biased, as presented in the paper, is 0 examples for one, and none for the other. What would happen if we had 2 examples for one and 4 for the other etc?
* This is not really a weakness of the paper, but rather talks about the fast pace of LLM growth. By the time I read the paper, GPT-3 is already not the best Open-AI model. Additionally, there exist models like instructGPT which allows different types of prompt engineering to circumvent such problems. A more complete work, while still in the scope of this workshop, could compare more models, more ways to define piece-wise functions etc.

**Suggested Changes:**

Figure 1 should also include OPT/GPT3 without examples, similar to what was shown in Figure 3.

In Appendix D.2, I assume the authors meant to write "much stronger inverse scaling when *biased* examples are provided."

---

### Author Response · Authors · 2023-06-01
**Submitting a Revised Manuscript and Opting-In to Archival**

Hello Tiny Papers Reviewers,

Thank you for the helpful feedback on our work! We have incorporated many of the provided suggestions in our revised submission.

We would like to opt-in to the archival of our manuscript.

All the best,
Paper171 Authors

---

### Comment · Area_Chair_fW8y · 2023-06-06
**Final meta-review: Invite to archive**

This work meets the threshold for archival, contents the URM statement and is deanonymized

---

### Meta-Review · Area_Chair_fW8y · 2023-04-04

**Recommendation:** Invite to present
**Confidence:** 4

**Metareview:**

The authors show that when LLMs are given biased examples on a type of function evaluation task they perform worse with more parameters. Conversely if unbiased example are provided, more parameters give more performance. The reviewers agree the paper is well written and the conclusions are justified but would like to see more experiments/analysis of the current experiments in the main body of the text.

Pros:
* The paper is well written and very clear
* The results support their claims.
* Two different LLMs are tested, both supporting their conclusions.

Cons:
* The reviews would like to see more "types" of biases: i.e. not just all or nothing but also where, for example, there is a 4:2 split, or no examples at all.
* Although the trend is clear it is not clear what is causing it and there is no intuition for why this should be the case. More tests, a deeper literature review and some educated speculations would be nice here.

The more confident reviewer assess this paper as Clear, Correct, and Reproducible (CCR), a judgement I agree with. Given the clarity or the paper and relatively positive review I judge this as "Recommend to present"

**Summary:**

The authors show that when LLMs are given biased examples on a type of function evaluation task they perform worse with more parameters. Conversely if unbiased example are provided, more parameters give more performance. The reviewers agree the paper is well written and the conclusions are justified but would like to see more experiments/analysis of the current experiments in the main body of the text.

**Comments And Feedback To The Authors:**

* It would help to show error bars in figure 1.

**Reason For Not Giving A Higher Recommendation:**

It is not clear _why_ the result shown is true. As reviewer says: "It is unclear whether the results are due to the model's misunderstanding of the task definition or failure to understand which function to use.". Without attempts to address this it is hard to know the impact of this interesting and well justified result.

**Reason For Not Giving A Lower Recommendation:**

It's a well written, clear paper. Both reviewers only had minor issues with its content and presentation.

---

### Decision · Program_Chairs · 2023-04-07

Invite to present